# Prevalence of anemia and its associated factors among adolescent girls on Weekly Iron Folic Acid supplementation (WIFAS) implemented and non-implemented schools at Tokha municipality, Kathmandu

Anita Khanal[1]*, Rajan Paudel[1,2], Chetan Nidhi Wagle[3], Shantos Subedee[4], Pranil Man Singh Pradhan[5,6]

1 Central Department of Public Health, Institute of Medicine, Tribhuvan University, Kathmandu, Nepal, 2 Unit of Health Sciences, Faculty of Social Sciences, Tampere University, Tampere, Finland, 3 National Health Training Center, Ministry of Health and Population, Teku, Kathmandu, Nepal, 4 Nepal Public Health Laboratory, Ministry of Health and Population, Teku, Kathmandu, Nepal, 5 Department of Community Medicine and Public Health, Institute of Medicine, Tribhuvan University, Kathmandu, Nepal, 6 Department of Global Health and Population, Harvard T.H. Chan School of Public Health, Boston, Massachusetts, United States of America

* phoanita@gmail.com

## Abstract

Anemia, the prevailing nutritional disorder worldwide, exerts its greatest impact on individuals in developing nations, exhibiting a heightened prevalence among adolescents. There is a window of opportunity for children to improve their nutritional status during their adolescence period. Weekly iron-folic acid supplementation is the preventive measure to break the intergenerational cycle of anemia. The correlation between the consumption of iron folic acid supplements and the prevention of anemia has been firmly established by numerous studies, revealing a statistically significant association. Therefore, this study was undertaken with the aim to assess the difference in the prevalence of anemia and its associated factors among adolescent girls in Weekly Iron Folic Acid Supplementation (WIFAS) implemented and non-implemented schools at Tokha Municipality, Kathmandu. A cross-sectional study was carried out among 602 adolescent girls from grade six to ten equally divided into WIFAS implemented and WIFAS non-implemented schools. Hemoglobin estimation was done using the HemoCue Hb 301 method. The descriptive, univariate and multivariable logistic regression analysis was done using SPSS version 25 to identify a significant association between anemia and its associated variables at p-value<0.05. The overall prevalence of anemia among adolescent girls was found to be 17.4% out of which, 14% and 20.9% from WIFAS-implemented and non-implemented schools respectively. The factors like type of school, fathers' education and dietary diversity were found to be statistically significant with anemia during multivariable logistic regression analysis. Given the findings of this research, proper counseling and promotion of utilization of WIFAS in order to decrease the prevalence of anemia among adolescent girls are recommended.

**Data Availability Statement:** All relevant data are within the paper and its Supporting Information files.

**Funding:** The authors received no specific funding for this work.

**Competing interests:** The authors have declared that no competing interests exist.

## Introduction

Adolescence, as defined by World Health Organization (WHO), is the transitional stage between childhood and adulthood, lasting from the age of ten to nineteen. This stage of life marked by rapid physical, cognitive, and emotional growth, lays the foundational bricks for a lifetime of good health and well-being [1]. A wide range of functional effects across the life course may result from anemia in adolescence, including poor academic performance, productivity loss and decreased present and future reproductive health for those who conceive [2]. Intermittent iron and folic acid supplementation is advised for menstruating women living in anemic areas to improve hemoglobin concentration and iron status in populations where the prevalence of anemia among non-pregnant women of reproductive age is 20% or higher [3, 4].

In 2019, there were 29.9% of women worldwide anemic, with the highest prevalence still being found in South-East Asia, where it is over 35% [5]. Anemia is estimated to affect 40% of all children between the ages of 6 and 59 months, 37% of pregnant women, and 30% of women between the ages of 15 and 49 worldwide [6]. In Nepal, anemia was more common among women aged 15 to 49, where it rose from 36% in 2006 to 41% in 2016, then fell to 34% in 2022. Nevertheless, the trend has not always been downward [7]. The time frame during pregnancy is too brief for women to build up enough iron reserves to meet the demands of the developing fetus. Anemic adolescent girls are more likely to give birth prematurely and to babies with low birth weight [8]. The weekly administration of "Iron Folic Acid" supplements is a potentially effective treatment for iron deficiency anemia, especially in adolescent girls [9].

Prior to the intervention, the occurrence of anemia among adolescent girls was 79.5%. However, through the implementation of a closely monitored administration of weekly iron-folic acid tablets and nutritional guidance, this figure declined to 58% [10]. Following the year-long intervention of weekly Iron and Folic Acid Supplementation (WIFAS), the initial anemia prevalence rate of 38% among adolescents dropped to 26% [11]. Iron and Folic Acid Supplementation (IFA) supplementation on adolescents' hemoglobin concentration has been shown to have positive effects in recent studies from Ghana, India, and Nepal [9, 12, 13]. According to a systematic review and meta-analysis conducted in Bangladesh from 1997 to 2019, iron supplements and foods enriched with micronutrients can be used to treat anemia in children between the ages of 6 months to 19 years [14].

WIFAS is a nationwide program implemented in Nepal since the 2015/16 as a public health intervention to prevent iron deficiency anemia for adolescent girls. It is typically offered through schools for in-school adolescent girls and from health facilities for out-of-school adolescent girls and constitutes of 60 mg elemental iron and 400 μg Folic acid. Following three months (13 weeks) of supplementation, there should be three months (13 weeks) of no supplementation before supplementation is to resume where each adolescent girl gets 26 IFA tablets total per year [15].

## Materials and methods

### Study setting, design, and population

This was a cross-sectional survey conducted in all WIFAS implemented schools and non-implemented schools of Tokha Municipality, Kathmandu. Tokha is an ancient historic city of Kathmandu valley in Bagmati province, which was formed in 2014 by merging five villages named as Dhapasi, Jhor Mahankal, Gongabhu, Chandeswori and Saraswoti. The total geographic area is 16.19 sq.km with total household of 32500 and total population is 149000. Through a self-administered questionnaire, a quantitative research methodology was used for

this study. The study population was adolescent girls aged 10–19 years from grade six to ten of Tokha municipality. In total, sixteen schools were selected for the study purpose. There were altogether eight WIFAS implemented schools in Tokha municipality and all were taken for study. Whereas eight WIFAS non-implemented schools were selected randomly on the basis of similar socioeconomic characteristics of respondents from the same ward of WIFAS implemented schools to ensure comparability. All the descriptive information about schools were obtained from the education section of the Tokha municipality.

## Sample size calculation and sampling technique

The sample size was calculated using the difference between two proportion formula as reported by a similar study conducted in Mangalore, Karnataka in India in 2022 [16]. The sample size was calculated using the formula of difference between two proportion.

$$n = (Z_{(1-\alpha/2)} + Z_{(1-\beta)})^2 * p_1(1-p_1) + p_2(1-p_2)/(p_1 - p_2)^2$$

where,

proportion of outcome from WIFAS non-implemented school ($p_1$) = 64.4% = 0.644
proportion of outcome from WIFAS implemented school ($p_2$) = 51% = 0.51
power (1-β) = 80% = 0.8
$Z_{alpha}$ value = 1.96 at 95% Confidence interval
$Z_{beta}$ value = 0.84 at corresponding 80% power
Therefore, putting the respective values,
The sample size for WIFAS implemented school ($n_1$) = 188.16
The sample size for WIFAS non implemented school ($n_2$) = 188.16
After calculation, the minimum sample size required was 376. After adjusting for the design effect of 1.5 and assuming a non-response rate of 10%, the final sample of 602 was calculated. The design effect of 1.5 was selected on the basis of a national level Nepal Demographic Health Survey (NDHS,2022) conducted. With the assumption of things like non-consent, non-participation and incomplete response 10% non-response rate was chosen. Its potential implications were that if the assumed non-response rate deviated from 10%, it could impact on generalizability of study findings and lead to bias in the results. The functional form of regression model:
y = log(p/(1-p)) = β0 + β1*x1 + . . . + βk*xk = Bt. X, for specific values of x
A list of adolescent girl students was obtained from each school from grade six to ten before the data collection. The number of clusters was determined by calculating the sum of the cumulative frequency of adolescent girls in each class using the Probability Proportional to Size (PPS) technique. This method gave each member of the population a probability of inclusion in the sample that was directly proportional to the sample's size. A random number generator was used to select population units and higher inclusion probabilities increased the likelihood of selecting units, while lower inclusion probabilities increased the likelihood of selecting smaller units, producing more precise and representative research findings. The class was considered a cluster. After the selection of the cluster, adolescent girls were chosen from each cluster for the hemoglobin test and asked to fill out the self-administered questionnaire.

## Data collection tools and procedure

The self-administered questionnaire was used for data collection. To ensure data quality, already validated tool was used. Q-Q plot displayed the data set was normally distributed. The completeness, consistency and timeliness of the tool was ensured by the research supervisors and experts in the field of nutrition. A well trained and licensed lab technician withdrew blood

from the respondents for hemoglobin estimation. Each respondent's assent and their parent's consent was taken before the blood was drawn for estimation. To draw blood from the finger, a sterile lancet was used. Blood was collected on a microcuvette after setting to zero and then inserted into the HemoCue device. The observed hemoglobin level of each respondent was noted on the questionnaire and communicated to them. Data collection was done by the researcher herself.

A standard questionnaire from the Adolescent Nutrition Health Survey 2014, Nepal was used to find the socio-demographic information. The knowledge on anemia, knowledge on WIFAS was adopted from a published article on Journal of nutrition and metabolism by SD Dubik et.al 2019. The dietary diversity questionnaire was adopted from FFQ, FAO, 2016. HemoCue Hb301 system, an effective tool that required few drops of blood from participant for hemoglobin estimation as it was diagnostically accurate and recommended by WHO to diagnose anemia. The researcher and the lab technician themselves measured their Hb level using the system to ensure the accuracy of the device.

The pretesting of the tool was done in 10% of sample size (n = 60) in one WIFAS implemented, and one non implemented school nearby Tarkeshwor municipality, Kathmandu to ensure the reliability of the tool. The pretesting helped to improve the respondents to interpret the questions correctly. On the basis of respondent debriefing, brief content orientation on anemia, WIFAS and skip patterns of questions were provided in each school before starting the data collection procedure in order to avoid biased answers and lead to improved data quality. The flow and sequence of the questions were changed with the guidance of the research supervisor.

A comprehensive informed consent form was created with all the required information regarding the study design, methodology, purpose, potential risks and benefits and the right of respondents to withdraw at any time. The parents were asked to sign the form only if they allow their child to participate in the study. The objectives of the study and instruction to fill the questionnaire was clearly stated to the participants

The knowledge of anemia and the knowledge of WIFAS was calculated by adding all relevant 19 and 15 knowledge items questions respectively. A correct answer for each item was scored as "1" and incorrect answer was scored as "0". The median mark obtained on knowledge of anemia was six whereas the median mark obtained on knowledge on WIFAS was five. The knowledge on anemia and WIFAS was considered good if the score was exactly equal to median and above. Similarly, the knowledge on anemia and WIFAS was considered poor if the score was below the median. Minimum dietary diversity was used to gauge dietary diversity practices, and adequate intake was defined as consuming at least five different food groups out of a possible ten in the previous 24 hours as defined by FFQ, FAO, 2016 (Table 1).

## Data analysis

First, the gathered data was manually compiled and verified. After that, EpiData 3.1 was used to code and enter the data. Exporting the data into SPSS (Statistical Package for Social Science) version 25 allowed for data analysis. Bivariate and multivariable logistic regression analysis was carried out to identify the association between anemia and independent variables.

Descriptive statistics (including means, standard deviations, frequencies and percentage) was calculated for the socio-demographic variables, knowledge related variables and dietary diversity. A Chi-square test was done for bivariate analysis to assess associations between the dependent variable (prevalence of anemia) and independent variables. In the test of multi-collinearity, none of the variables had tolerance $<0.1$ and Variance Inflation Factor was found to be $<10$ which ensured that there is no relationship or interdependence between independent

**Table 1. Summary of study variables.**

| Variables | Definitions of Variables | Measurements |
|---|---|---|
| Anemia | Defined by WHO, 2011 | Hemoglobin level ≥12 (non-anemic)<br>Hemoglobin level <12g/dl (anemic)<br>Mild anemia (11 to 11.9 gm/dl)<br>Moderate anemia (8 to 10.9gm/dl)<br>Severe anemia (< 8gm/dl) |
| Age | Age of the student in completed years at the time of the study | 10–14 years (early adolescent)<br>15–19 years (late adolescent) |
| Type of family | Type of family based on the composition of family members | Nuclear<br>Joint |
| Father education | The level of education attained by the respondent's father | No education<br>Basic education (grade one to eight)<br>Secondary education (grade nine to twelve)<br>More than secondary education (grade thirteen and above) |
| Mother education | The level of education attained by the respondent's mother | No education<br>Basic education (grade one to eight)<br>Secondary education (grade nine to twelve)<br>More than secondary education (grade thirteen and above) |
| Father occupation | The type of occupation of respondent's father | Agriculture, Housemaker, Government Service, Private business, Foreign Employment, Daily wages and Others |
| Mother occupation | The type of occupation of respondent's mother | Agriculture, Housemaker, Government Service, Private business, Foreign Employment, Daily wages and Others |
| Knowledge on anemia and WIFAS | It was calculated by adding up all related questionable items. Each item received a score of "1" for a correct response and "0" for an incorrect response | Scored median and above the median value was classified as having good WIFAS and anemia knowledge, scored below the median value was classified as having poor WIFAS and anemia knowledge |
| Dietary Diversity | FAO, 2016 | Adequate dietary diversity (consumed ≥ 5 food groups)<br>Inadequate dietary diversity (consumed 4 and less food groups) |

variables themselves and were eligible for multivariable logistic regression analysis. The p-value was less than 0.05 which meant that the full model was statistically significant and predicted the dependent variable. The strength of association was measured using adjusted odds ratio (AOR) at 95% confidence interval.

## Ethical approval and consent

Ethical approval was taken from Institutional Review Committee of Institute of Medicine (IoM), Nepal [Ref: 265(6–11) E2 2079/080]. Formal permission was taken from Tokha Municipality Office and concerned school authority and consent and assent was taken from the participants and parents before collecting data. Personal identity information was not included in the questionnaires and to maintain the anonymity of respondents, unique codes were used during the entire period of study. Respondents were allowed to fully participate on their wishes and could withdraw anytime and their choice of not participating in the study was respected. Data collection, handling, and analysis were done solely by the principal investigator, and confidentiality were maintained during every step of the research. The data was collected after receiving the ethical approval from IRC from December 18, 2022 to January 25, 2023.

The respondents were clearly stated and counselled about the blood draw procedure from the tip of her finger, timing and hemoglobin level and status of anemia. In case of severity, she

will be referred to nearby health facility with proper counselling and nutrition education on essence and regular intake of iron rich foods and weekly iron folic acid supplementation. The respondents were assured that they will not have any financial burden.

## Results

### Socio-demographic characteristics of study participants

Only 29.2% of respondents in the non-WIFAS implemented group were late adolescent compared to 51.2% in the WIFAS implemented school. In contrast, the WIFAS implemented school had 48.8% early adolescent compared to 70.8% in the non-WIFAS school. In comparison to the non-WIFAS school, the WIFAS implemented school has a slightly older mean age (14.48 years vs. 13.77 years). Both the implemented and non-implemented schools had comparatively higher number of participants from grade six to eight. Overall, the distribution of first menstrual age was slightly different between the two schools. There was no substantial difference between the mean age of menarche.

The advantaged group (Brahmin/Chhetri) made up 32.6% of participants in the "WIFAS implemented" group, while the non-advantaged group (Others) made up 67.4%. The others category included Janajati, Madeshi, Dalit and Muslim. When compared to the non-WIFAS schools, the WIFAS implemented schools have a higher proportion of respondents from the "Janajati" ethnicity (55.8%). In terms of religion, the non WIFAS implemented school had more "Hindu" people than the WIFAS implemented (70.1% vs. 55.1%). With only minor variations, respondents in "Extended" and "Nuclear" family were distributed similarly across the two groups. In compared to the non-WIFAS, the WIFAS implemented schools had a higher percentage of respondents with education levels "Below secondary" in both the mother's and father's education categories. More people in the non-WIFAS schools had parents who were "Secondary and above" educated. In both schools, mothers and fathers of the respondents were mostly involved in the non-agricultural activities. There was 100% response rate of the respondents (Table 2).

### Anemia, WIFAS and dietary diversity

In the study, the overall prevalence of anemia was found to be 17.4%. Specifically, 20.9% adolescent girls from WIFAS non-implemented and 14% from WIFAS implemented schools were anemic. In WIFAS implemented schools, 8% had mild anemia, 5.7%had moderate anemia, and 0.3%had severe anemia. In contrast, 15.9% had mild anemia, and 5% had moderate anemia in WIFAS non-implemented schools.

Hemoglobin levels among adolescent school girls exhibited a range of 6.70g/dl to 18.0g/dl, with a mean value of 13.30± (SD 1.42). Average hemoglobin level was determined to be 13.29 g/dl in WIFAS non-implemented schools and 13.32 g/dl in WIFAS implemented schools. Hemoglobin levels varied from 8.4g/dl to 16.6g/dl in WIFAS non- implemented school and 6.7g/dl to 18g/dl in WIFAS implemented schools. A Q-Q plot (quantile-quantile plot) was used to ensure data quality and to compare the probability distributions of hemoglobin level between WIFAS and non-WIFAS schools, which indicated that the data was normally distributed because the nearly straight line closely followed the diagonal line.

About 59.1% and 54.2% of the respondents from WIFAS implemented school and non-implemented schools respectively had good knowledge on anemia. About 78.4% and 46.5% of the respondents from WIFAS implemented and non-implemented schools were considered having good knowledge on WIFAS. This study found that adolescent girls from non-WIFAS implemented schools had less inadequate food diversity than those from WIFAS-implemented schools, which represented 25.2%, and 29.2% respectively. It implied that adolescents from

**Table 2. Socio-demographic characteristics of the adolescents (n = 602).**

| Characteristics | WIFAS implemented (n = 301) | Non-WIFAS implemented (n = 301) |
|---|---|---|
| | n (%) | n (%) |
| **Age** | | |
| 15–19 years (late adolescent) | 154 (51.2) | 88 (29.2) |
| 10–14 years (early adolescent) | 147 (48.8) | 213 (70.8) |
| **Mean age ± SD** | 14.48(±1.45) years | 13.77(±1.45) years |
| **Grade** | | |
| Nine-Ten | 144 (47.8) | 85 (28.2) |
| Six- Eight | 157 (52.2) | 216 (71.8) |
| **First Age of menstruation (years) (n = 523)** | | |
| Eleven | 45 (16.8) | 59 (23.2) |
| Twelve | 110 (41) | 113 (44.3 |
| Thirteen | 72 (26.9) | 62 (24.3) |
| Fourteen | 41 (15.3) | 21 (8.2) |
| **Mean age of menarche** | 12.40(±0.94) years | 12.17(±0.88) years |
| **Ethnicity** | | |
| Advantaged group(Brahmin/Chhetri) | 98 (32.6) | 145 (48.2) |
| Non-advantaged group (Others) | 203 (67.4) | 156 (51.8) |
| **Religion** | | |
| Non Hindu | 135 (44.9) | 90 (29.9) |
| Hindu | 166 (55.1) | 211 (70.1) |
| **Family type** | | |
| Extended | 112 (37.2) | 102 (33.9) |
| Nuclear | 189 (62.8) | 199 (66.1) |
| **Mother's education** | | |
| Below secondary | 242 (80.4) | 147 (48.8) |
| Secondary and above | 59 (19.6) | 154 (51.2) |
| **Father's education** | | |
| Below secondary | 175 (58.1) | 86 (28.6) |
| Secondary and above | 126 (41.9) | 215 (71.4) |
| **Father's occupation** | | |
| Agriculture | 64 (21.3) | 16 (5.3) |
| Non agriculture | 237 (78.7) | 285 (94.7) |
| **Mother's occupation** | | |
| Agriculture | 31 (10.3) | 15 (5) |
| Non agriculture | 270 (89.7) | 286 (95) |

non-WIFAS implemented school had adequate food diversity in compared to WIFAS implemented schools. The mean food group consumption was 5.35 with SD ± 1.39.

**Association of socio-demographic characteristics, knowledge of anemia and WIFAS and dietary diversity with anemia.** Anemia was significantly correlated with adolescent age, grade, type of school, mother's occupation and father's education. The odds of being anemic were 1.58 times higher for late adolescents compared to early adolescent. The likelihood of adolescents in WIFAS non-implemented schools being anemic was 1.6 times higher than it was for adolescents in WIFAS implemented schools. The study findings indicated that there was a notable association between grade level and the odds of anemia. The odds of anemia were found to be 1.7 times higher in grade ranging from nine to ten, in comparison to grade six to eight. Furthermore, respondents whose mothers had below secondary education were

3.68 times more likely to experience anemia (OR 3.68, 95% CI 2.08–6.40) compared to respondents whose mothers had attained secondary education or higher. Additionally, when examining the cross-tabulation, the education level of the father and the occupation of the mother showed significant associations with anemia.

The odds of anemia among adolescent girls with poor knowledge on anemia was 1.5 times higher in compared to odds of anemia among adolescents with good knowledge on anemia. Similarly, the WIFAS non-implemented schools was found to be 1.6 times higher odds of anemic in compared to WIFAS implemented schools. Similarly, the odds of anemia among adolescents with poor WIFAS knowledge was 1.6 times higher in compared to odds of anemia in adolescents with good WIFAS knowledge. There was a significant association between type of school and WIFAS knowledge with the anemia. Regarding the dietary diversity, the study showed that participants with inadequate dietary diversity were 13.80 times to be anemic (OR 13.80 at 95% CI 8.411–22.61) in compared to participants with adequate dietary diversity. However, the study did not reveal a significant relationship between the implementation of the Weekly Iron-Folic Acid Supplementation (WIFAS) program and dietary diversity (Table 3).

## Multivariable logistic regression analysis

Table 4 displayed the findings of multivariable logistic regression analysis for the key predictors of the prevalence of anemia. The factors that persisted in the final model were age, type of school, classes, mother's education, father's education, occupation of mother, knowledge on anemia, knowledge on WIFAS and dietary diversity. In the final model, the WIFAS non-implemented schools (AOR: 3.1, 95% CI 1.7–5.7), lower level of education of father (AOR 3.7, 95% CI 1.8–7.5) and inadequate dietary diversity (AOR 12.1, 95% CI 6.9–20.8) were found to be significantly associated with anemia among school going adolescent girls.

**Limitations of the study.** The cross-sectional study design used in the study couldn't measure the causal inference as the exposure and outcome were examined at the same time.

In the study, the impact of deworming on anemia prevalence couldn't be assessed.

The level of serum ferritin couldn't be assessed to measure iron storage.

Long term impact of weekly iron folic acid supplementation intervention on student's knowledge, compliance and anemia status couldn't be assessed.

There was a chance for recall bias because questions about food frequency were asked retrospectively.

## Discussion

Anemia is a public health problem in Nepal. The intergenerational cycle of malnutrition and anemia can be broken by treating adolescent anemia. Anemia can lead to reduced work productivity, increased health care costs, interconnected health issues and loss of human capital. According to estimates, if US\$1 were spent on reducing women's anemia, US\$12 in potential economic returns could result [17]. Therefore, the Weekly Iron and Folic Acid Supplementation (WIFAS) Program is a low cost preventive measure to help adolescent girls who are planning to become pregnant avoid anemia [11]. According to the current study, anemia was prevalent among adolescent girl students from grade sixth to tenth accounting 17.4%. The prevalence of anemia was higher in WIFAS non-implemented schools (20.9%) in compared to WIFAS-implemented schools (14%). The differences in the prevalence of anemia among WIFAS implemented and non-implemented schools were significant. These differences in the anemia rates is due to the implementation of WIFAS program in government schools, where the girls were directly benefited from the tablets that fulfil the iron requirements of the body.

**Table 3. Association of socio-demographic characteristics, knowledge of anemia and WIFAS and dietary diversity with anemia (n = 602).**

| Characteristics | Anemic | Non-anemic | p-value | OR | 95% CI |
|---|---|---|---|---|---|
| | n (%) | n (%) | | | |
| **Age** | | | | | |
| 15–19 | 52 (21.5) | 190 (78.5) | 0.032 | 1.6 | 1.1–2.4 |
| 10–14 | 53(14.7) | 307 (85.3) | | Ref. | |
| **Type of school** | | | | | |
| Non-Implemented | 63 (20.9) | 238 (79.1) | 0.024 | 1.6 | 1.1–2.5 |
| WIFAS implemented | 42 (14) | 259 (86) | | Ref. | |
| **Grade** | | | | | |
| 9 –10 | 51(22.3) | 178 (77.7) | 0.014 | 1.7 | 1.1–2.5 |
| 6–8 | 54 (14.5) | 319 (85.5) | | Ref. | |
| **Mothers' education** | | | | | |
| Below secondary | 89 (22.9) | 300 (77.1) | <0.001 | 3.7 | 2.1–6.4 |
| Secondary and above | 16 (7.5) | 197 (92.5) | | Ref. | |
| **Father's education** | | | | | |
| Below Secondary | 78 (29.9) | 183 (70.1) | <0.001 | 4.9 | 3.1–7.9 |
| Secondary and above | 27(7.9) | 314 (92.1) | | Ref. | |
| **Religion** | | | | | |
| Non-Hindu | 35 (15.6) | 190 (84.4) | 0.346 | 0.8 | 0.6–1.3 |
| Hindu | 70 (18.6) | 307 (81.4) | | Ref | |
| **Type of family** | | | | | |
| Extended | 42 (19.6) | 172 (80.4) | 0.294 | 1.2 | 0.8–1.9 |
| Nuclear | 63 (16.2) | 325 (83.8) | | Ref. | |
| **Mother's occupation** | | | | | |
| Agriculture | 15 (32.6) | 31(67.4) | 0.005 | 2.5 | 1.3–4.8 |
| Non-agriculture | 90 (16.2) | 466 (83.8) | | Ref. | |
| **Father's occupation** | | | | | |
| Agriculture | 16 (20) | 64 (80) | 0.517 | 1.2 | 0.7–2.2 |
| Non-agriculture | 89 (17) | 433 (83) | | Ref. | |
| **Knowledge on anemia** | | | | | |
| Poor knowledge | 55 (21.1) | 206 (789) | 0.040 | 1.554 | 1.2–2.4 |
| Good knowledge | 50 (14.7) | 291 (85.3) | | Ref. | |
| **Knowledge on WIFAS** | | | | | |
| Poor knowledge | 49 (21.7) | 177 (78.3) | 0.034 | 1.582 | 1.1–2.4 |
| Good knowledge | 56 (14.9) | 320 (85.1) | | Ref. | |
| **Dietary diversity** | | | | | |
| Inadequate food diversity | 78 (47.6) | 86 (52.4) | <0.001 | 13.806 | 8.4–22.6 |
| Adequate food diversity | 27(6.2) | 411 (93.8) | | Ref. | |

This is Nepal's first comprehensive study looking into adolescent anemia after the implementation of WIFAS program by government of Nepal in WIFAS implemented and non-implemented schools at Tokha municipality. The results give a national estimate of adolescent anemia, which can be used as a benchmark to assess the effectiveness of the Weekly Iron Folic Acid Supplementation nutrition programs that are currently in place and to calculate the National Burden of Disease for Nepal. Additionally, in the context of Nepal at the national level, this finding could sheds light on the associated factors that contribute to adolescent girls' anemia.

**Table 4. Adjusted relationship of independent variables with the anemia (n = 602).**

| Characteristics | Bivariate analysis | | Multivariable logistic regression analysis | |
|---|---|---|---|---|
| | COR (95%CI) | P value | AOR (95%CI) | P value |
| **Age** | | | | |
| 15–19 | 1.6 (1.04–2.4) | 0.032 | 0.8 (0.4–1.7) | 0.600 |
| 10–14 | Ref. | | Ref. | |
| **Type of school** | | | | |
| Non-implemented | 1.6 (1.06–2.5) | 0.024 | 3.1 (1.7–5.7) | <0.001 |
| WIFAS implemented | Ref. | | Ref. | |
| **Grade** | 1.7 (1.1–2.5) | 0.014 | 2.1 (0.9–4.6) | 0.051 |
| 9–10 | | | | |
| 6–8 | Ref. | | Ref. | |
| **Mother's education** | | | | |
| Below secondary | 3.7 (2.1–6.4) | <0.001 | 1.4 (0.6–3.1) | 0.414 |
| Secondary and above | Ref. | | Ref. | |
| **Father's education** | | | | |
| Below secondary | 4.9 (3.1–7.9) | <0.001 | 3.7 (1.8–7.5) | <0.001 |
| Secondary and above | Ref. | | Ref. | |
| **Occupation of mother** | | | | |
| Agriculture | 2.5 (1.3–4.8) | 0.005 | 0.9 (0.4–2.3) | 0.895 |
| Non-agriculture | Ref. | | Ref. | |
| **Knowledge on anemia** | | | | |
| Poor knowledge | 1.6 (1.02–2.4) | 0.04 | 1.5 (0.9–2.6) | 0.124 |
| Good knowledge | Ref. | | Ref. | |
| **Knowledge on WIFAS** | | | | |
| Poor knowledge | 1.6 (1.0–2.4) | 0.034 | 1.2 (0.7–2.2) | 0.436 |
| Good knowledge | Ref. | | Ref. | |
| **Dietary diversity** | | | | |
| Inadequate | 13.8 (8.4–22.6) | <0.001 | 12.1 (6.9–20.8) | <0.001 |
| Adequate | Ref. | | Ref. | |

Anemia prevalence among adolescent girls decreased from 79.5% to 58%, according to a study conducted by Shobha P. Shah, demonstrating the effectiveness of WIFAS in reducing anemia [10]. Likely, in our study, WIFAS implemented school girls had a lower prevalence of anemia than girls from non-implemented schools. Our result was in line with studies on young people from Etumanor panchayat and Nepal found that the prevalence of anemia was respectively 21% and 31%, which is relatively low [3, 13]. Different studies have revealed the effectiveness of WIFAS in reducing the prevalence of anemia conducted throughout the countries [10–12, 16].

In the study conducted there was a greater incidence of anemia in late adolescents (15 to 19) years as compared to that of early adolescents (10 to 14) years. These findings in contrast to the findings of NDHS 2022 that has focused anemia prevalence decreases as people age, where 39.4% were anemic of 15–19 years as compared to 34.5% of 20–29 years [7]. However, the findings of this study were in alignment with the findings studied by Wangaskar, et al., where late adolescents were more likely to have anemia than early adolescents [18]. This might be due to inconsistent use of IFA tablets and increased menstrual blood loss among late adolescents.

Different studies have revealed that there is positive association between intake of WIFAS and hemoglobin level [11–13]. The overall results indicated that anemia prevalence decreased

from 79.5% to 58% among adolescent girls with the average increase in hemoglobin was 1.3 g/dl for adolescent girls [10]. Similarly, the overall prevalence of anemia decreased from 73.3% to 25.4% in 4 years [19]. At the beginning of the intervention, the prevalence of anemia was 38% by the end of the year, it had dropped to 26%. The average hemoglobin level increased to 0.37 g/dl in a year interventions [11]. Increases in hemoglobin levels vary depending on how frequently and for how long WIFAS is used.

Regarding the socio demographic characteristics, the mother's and father's education were found to have significant association with anemia. Similar findings were resulted in the study conducted in India and Ghana [9, 20]. The educated parents were more likely to discuss on measures on anemia prevention, eating of iron rich foods and adopting healthy practices. This would help to prevent anemia since early pre pregnancy period. The mean age of menarche was 12.29(±0.92) years in this study conducted as in line with study by Subramanian M et al. [21]. The WIFAS implemented school had a slightly older mean age (14.48 years) as compared to non-implemented schools (13.77 years). However, as it can be challenging for adolescent girls to consume enough iron to make up for the menstrual iron losses, menstrual blood loss can significantly contribute to iron depletion. This is crucial because nearly all the adolescent girls who participated in the study had experienced menarche, which can further affect their iron status.

The study manifested that the majority (84.7%) of the adolescent girls have good knowledge of anemia and Iron and Folic Acid. The study showed 38.9% of the respondents having Hb levels less than 12g/dl, and the prevalence of anemia according to WHO was "medium" [22]. This finding were almost similar to this study where about 83% of adolescent girls were non anemic and more than two third of the respondents had good knowledge on anemia, this might be the reason to intake iron rich foods and take protective measures to prevent anemia. In this study, the adolescent girls were found to be motivated to take the IFA tablet for the prevention of anemia due to teacher advice.

The adolescent girl's knowledge on anemia and WIFAS was found significant with prevalence of anemia and the WIFAS implemented school adolescent girls have comparatively good knowledge on anemia and WIFAS. The study conducted in Indonesia on the awareness on anemia and WIFAS among girls' adolescents found that the benefits of consuming WIFAS were boosting stamina, adding iron, better digestion, overcoming premenstrual syndrome and adding red blood cells. But the girls claimed that they decided not to consume IFA during exams due to the negative effects [23]. However, in this study, WIFAS helped to increase the blood level was reported by majority of the respondents. The knowledge on foods that inhibit iron absorption was lower. This finding could be as a result of the socio-cultural environment and the availability of foods that had an impact on the knowledge of girls and parents regarding foods and drinks that inhibit iron absorption.

Regarding the prevalence of dietary diversity practice, of the total respondents (32.3%) reported adequate dietary practices. The adequacy of micronutrients is significantly correlated with dietary diversity. High dietary diversity is typically viewed as a beneficial aspect of a diet [24]. Similarly, in a study conducted in Ethiopia, adolescent girls with inadequate dietary diversity had two times (AOR = 2.1; 95% CI; 1.3, 3.5) higher risk of anemia than those with adequate dietary diversity [25]. In this study, participants with inadequate dietary diversity were 12.1 times odds ratio to be anemic (OR 12.1 at 95% CI 6.9–20.8) in compared to participants with adequate dietary diversity suggesting a significant relationship between adequate dietary diversity (at least five food groups) and lower prevalence of anemia. Low dietary intake of food groups high in nutrients, such as eggs, milk, and meat, might be the cause.

Higher MDD-W prevalence in a group of WRA serves as a proxy for higher micronutrient adequacy in a particular population [26]. Cereals were consumed universally and the mean

dietary diversity score in the study was five. Another predictor identified in the study was the type of school, results showed that adolescents enrolled in WIFAS implemented schools were more likely than those enrolled in non-implemented schools to have low dietary diversity scores [27]. This might be linked due to the different socioeconomic status of the family. This result was in line with research done in Jima Town [28]. However, in this study, the schools with lower dietary diversity were found to have lower prevalence of anemia in compared to schools with better dietary diversity that in fact have higher prevalence of anemia. Thereby, more investigation seems to be required to carry on adolescent nutrition and to ascertain how dietary habits and food beliefs impact adolescent anemia.

One of the eight essential, highly effective actions for enhancing adolescent nutrition identified by the WHO in the 2018 recommendations is weekly iron and folic acid supplementation (WIFAS). However, the Model Essential Medicines List (MEML) currently does not list WIFAS in the WHO-recommended formulation. In order to reduce anemia and prevent neural tube defects, WIFAS could be proposed to the MEML. With this inclusion, efforts to reduce anemia and neural tube defects globally could be accelerated in their implementation [29]. Lassi et al.'s recent review revealed that iron supplements could significantly raise adolescent hemoglobin concentration. To increase the amount and bioavailability of micronutrients in family diets, efforts should also be directed toward nutritional programs that encourage dietary diversity [30].

## Conclusion

The WIFAS-implemented schools in the study were government schools with a lower prevalence of anemia. However, the WIFAS-non-implemented schools were private schools, where the prevalence of anemia was higher, despite the fact that students in these schools had better access to diverse food due to their comparative better socioeconomic conditions. Hence, the WIFAS program, which provides weekly supplements of iron and folic acid significantly reduces anemia. Its supplementation administered at schools is effective in lowering anemia prevalence in schoolgirls. As per the multivariable logistic regression analysis, the type of school was found to be significant, where the odds of anemia were 3.1 times higher in WIFAS non-implemented schools in compared to WIFAS implemented schools. More, specifically, the present study presented that WIFAS implemented schoolgirls had a lower prevalence of anemia (14%) than girls in non-implemented schools (20.9%), suggesting that WIFAS played a significant role in reducing anemia prevalence.

The type of school, father's education and dietary diversity were the main factors in the study that were statistically associated with the prevalence of anemia. The promotion of proper utilization of the Weekly Iron and Folic Acid Supplementation (WIFAS) program is imperative. Informing and counseling adolescent girls along with their parents about the program's advantages in order to decrease the prevalence of anemia among adolescent girls are recommended. The study findings of WIFAS implementing school having lower anemia than non-implementing schools are obvious and reasonable and can be applied to other schools as well across the country.

### Implications of the findings

The prevalence of anemia was found lower among WIFAS implemented schools where there was provision of school heath nurse. Therefore, a school heath nurse has to be recruited in every government and non-government schools to motivate and counsel adolescent girls on the benefits and consumption of WIFAS.

The findings of the study can be utilized for the evidence based public health policy formulation addressing WIFAS non-implemented schools, community based schools and out of school adolescent girls, where they are at risk to anemia.

Effective communication and implementation of WIFAS program can lead to improved adolescent health and ultimately to societal health and wellbeing.

## Supporting information

**S1 File. Self-administered questionnaire.**
(DOCX)

**S1 Data. Data file.**
(SAV)

## Acknowledgments

I am grateful to Tribhuvan University, Central Department of Public Health for providing the Hemocue Hb 301 system kit for this study. I am indebted to Tokha Municipality for facilitating my data collection process. I would like to appreciate all the adolescent girls for their participation, their parents, schools' principals and school health nurses to complete the data collection.

## Author Contributions

**Conceptualization:** Anita Khanal, Rajan Paudel.

**Formal analysis:** Anita Khanal, Chetan Nidhi Wagle.

**Investigation:** Shantos Subedee.

**Methodology:** Anita Khanal, Chetan Nidhi Wagle, Shantos Subedee, Pranil Man Singh Pradhan.

**Resources:** Anita Khanal.

**Software:** Chetan Nidhi Wagle.

**Supervision:** Rajan Paudel, Pranil Man Singh Pradhan.

**Writing – original draft:** Anita Khanal.

**Writing – review & editing:** Anita Khanal, Rajan Paudel, Pranil Man Singh Pradhan.

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
