## [Decision Letter · Decision Letter 0]

6 Sep 2023

PGPH-D-23-01329

Prevalence of anemia and its associated factors among adolescent girls on Weekly Iron Folic Acid supplementation (WIFAS) implemented and non-implemented schools at Tokha municipality, Kathmandu

Dear Ms/Mrs.Khanal,

Thank you for submitting your manuscript to PLOS Global Public Health. After careful consideration, we feel that it has merit but does not fully meet PLOS Global Public Health’s publication criteria as it currently stands. Therefore, we invite you to submit a revised version of the manuscript that addresses the points raised during the review process.

We look forward to receiving your revised manuscript.

Kind regards,

Jitendra Kumar Singh, PhD

Academic Editor

Journal Requirements:

1. Please include a complete copy of PLOS’ questionnaire on inclusivity in global research in your revised manuscript. Our policy for research in this area aims to improve transparency in the reporting of research performed outside of researchers’ own country or community. The policy applies to researchers who have travelled to a different country to conduct research, research with Indigenous populations or their lands, and research on cultural artefacts. The questionnaire can also be requested at the journal’s discretion for any other submissions, even if these conditions are not met.  Please find more information on the policy and a link to download a blank copy of the questionnaire here: https://journals.plos.org/globalpublichealth/s/best-practices-in-research-reporting. Please upload a completed version of your questionnaire as Supporting Information when you resubmit your manuscript."

2. We have noticed that you have uploaded Supporting Information files, but you have not included a list of legends. Please add a full list of legends for your Supporting Information files after the references list. 

Additional Editor Comments (if provided):

This research addresses a significant topic and has the potential to contribute valuable insights to the scientific community; however, there are a few areas where we believe your manuscript could benefit from some improvements to enhance its overall quality and impact.

Study setting, design, and population

The description of the study settings gives helpful background for the readers. However, it would be helpful to have more detailed information, such as the number of schools that have selected for the study in WIFAS implemented and not implemented of Tokha municipality.

Sample size calculation and sampling technique

It's a worthy you highlighted the method used to calculate sample size and the reference to a comparable research for explanation. However, it is better to provide more details information about the parameters assumed for this study. Furthermore, it would be worthy to describe on what basis a design effect of 1.5 was decided. Authors mentioned the use of Probability Proportional to Size (PPS) technique for cluster selection. This seems confusion to the readers, therefore it would be helpful to provide a brief explanation of what PPS is and how it was applied in for study.

Discussion

Contextualize the Study: It would be constructive to begin discussions by providing a quick summary of the importance of addressing anemia among adolescent of Nepal. It may include information on the health and economic effects of anaemia, as well as why this study is important, and then emphasizing WIFAS's success in terms of elements of WIFAS that contribute to its effectiveness, and how does it compare to other interventions as you demonstrated the effectiveness of WIFAS in reducing anemia

Conclusion

These findings have strong policy implications for enhancing adolescent girl’s health. As a result, it is recommended that you discuss policy implications based on your results.

Reviewers' comments:

Reviewer's Responses to Questions

**Comments to the Author**

1. Does this manuscript meet PLOS Global Public Health’s publication criteria? Is the manuscript technically sound, and do the data support the conclusions? The manuscript must describe methodologically and ethically rigorous research with conclusions that are appropriately drawn based on the data presented.

Reviewer #1: Partly

Reviewer #2: Yes

Reviewer #3: Partly

2. Has the statistical analysis been performed appropriately and rigorously?

Reviewer #1: Yes

Reviewer #2: Yes

Reviewer #3: I don't know

3. Have the authors made all data underlying the findings in their manuscript fully available (please refer to the Data Availability Statement at the start of the manuscript PDF file)?

Reviewer #1: Yes

Reviewer #2: Yes

Reviewer #3: No

4. Is the manuscript presented in an intelligible fashion and written in standard English?

Reviewer #1: No

Reviewer #2: Yes

Reviewer #3: Yes

5. Review Comments to the Author

Reviewer #1: Thanks for the opportunity to review the paper titled- “Prevalence of anemia and its associated factors among adolescent girls on Weekly Iron Folic Acid supplementation (WIFAS) implemented and non-implemented schools at Tokha municipality, Kathmandu.” Although, the paper looks good, it needs to revise as follows-

1. Grammatical issues should have been revised in the whole manuscript.

2. Providing some more global statistics on anemia would make the introduction better. Moreover, condition of anemia in Nepal should have been included.

3. Mentioning the exact equation to calculate the sample size would be better.

4. In the prevalence section of the results, frequency of the participants should have been omitted and more focus should have been given to the percentage.

5. Results should have been described in past tense.

6. There is an inconsistency in the use of “non-implemented schools”. Author should correct all the “not implemented schools” to “non-implemented schools”.

7. Authors have mentioned both “multivariate analysis” and “multivariable logistic regression analysis”. Author should clarify about which one he/she used; As there is a difference between these 2 (two) analysis method.

8. “Limitation of the study” should have been mentioned in a separate section.

9. “In the study, the impact of deworming on anemia prevalence couldn’t be assessed. The level of serum ferritin couldn’t be assessed to measure iron storage”- This part of page 5 should have been included as the limitation of the study.

10. “The knowledge of anemia and the knowledge of WIFAS was calculated by adding all relevant 19 and 15 knowledge items questions respectively. A correct answer for each item was scored as “1” and incorrect answer was scored as “0”- This part of page 10 should have been included in the method section of the study.

11. “Analysis of association of socio-demographic, knowledge on anemia and WIFAS and dietary diversity with anemia”- This headline should be reworded as- “Association of socio-demographic characteristics, knowledge of anemia and WIFAS and dietary diversity with anemia”

Reviewer #2: 1. The data collection procedures adopted lacks details about how the questionnaires were distributed, collected, and managed. Information on data collection supervision, training of data collectors, and quality control measures would enhance the methodology's robustness. For instance, how were the students approached, informed about the study, and how was their consent obtained?

2. The knowledge on anemia, knowledge on WIFAS, dietary diversity and compliance-related questionnaire was adapted from different published articles. Which articles are these and what exactly did the authors adopt from each of the articles.

3. More information about the process of pretesting, findings from pretesting, and the nature of modifications made to the questionnaire after pretesting and how they improved the tool should be explained.

4. Did the questionnaire undergo any content validation or reliability testing? if yes explain the process and the outcome.

5. The authors should include the duration of the data collection period.

6. The section could provide a brief outline of the types of the specific statistical analyses conducted and the purpose of each.

7. The section mentions some limitations, but it could be more comprehensive. Addressing limitations associated with the study design, data collection, and analysis adds to the study's credibility.

8. While the non-response rate assumption of 10% is mentioned, it could be helpful to provide reasons for this assumption and acknowledge potential implications.

9. The authors mentioned obtaining consent from parents before the blood draw, but it doesn't elaborate on how parental consent was obtained. Clarifying this process would strengthen the ethical considerations.

10. What were the steps taken to ensure data quality, such as data validation checks or inter-rater reliability checks for the self-administered questionnaire?

11. What was the response rate?

12. What are the implications of your finding?

Reviewer #3: This paper reports on the prevalence on anemia in school-based populations with and without a Weekly Iron Folic Acid Supplementation (WIFAS) program in place. The authors find that among girls in schools with WIFAS, anemia rates are substantially lower than in the schools without (14% vs 20.9%).

This finding, as well as the complementary analysis of the additional factors associated with anemia will provide a valuable addition to the literature. However, with a substantial revision, I think it could be greatly strengthened, and I recommend major and minor changes below that I think the authors should undertake before this paper is ready for publication in a journal such as PGPH.

Major issues

Methods

I don’t understand your sample size calculation section (middle of page 7 of the pdf); I would expect that you need to include an assumed prevalence of anemia as well as an assumed effect size of WIFAS to conduct the power calculation.

Furthermore, it seems essential to include additional detail on how the eight schools were selected from the WIFAS and non-WIFAS groups. If they were selected uniformly at random, then your results are more likely to be generalizable than if they were selected in some non-random manner.

It is also important for the reader to know more about how the schools were selected to have or not have the WIFAS program in the first place. If this was through a randomization process it is much more generalizable than if it was selected purposefully in some manner (e.g. because these are schools where the students are known or suspected to have higher anemia prevalence).

I find the description of your sampling design once you have selected the schools unclear as well, and would request a revised description of this, too.

Results

I would like a report on the participation rate and amount of missing data. Please also describe how you handled missing data in your methods section.

I would like all of your Table 2 characteristics reported for the WIFAS and non-WIFAS schools separately, to see if these schools seem similar on these dimensions.

Supporting information

I think you should include the individual participant data (appropriately anonymized) as an additional supporting document.

Minor issues

Page 5, line 3: “the fond ages of ten to nineteen”, the word fond sound too informal to me

Page 5, line 3: “this is a remarkable stage of life”, this phrase sounds awkward to me

Page 5, line 7: reference 3 seems overly focused, with a title about girls in Central Kerela. Please include a more general reference here instead.

Page 5, line 10: double check if these references are what you intended; ref [4] seems like it might be intended for elsewhere.

Page 5, line 9 from bottom: is there an evidence synthesis that you can cite here, such as a systematic review and meta-analysis?

Page 5, final paragraph: this has a lot in common with the previous paragraph, see if you can merge them.

Page 5, line 3 from bottom: as an non-expert reader, I would like to know more about the state of WIFAS and its context. Has this been piloted extensively before? Has it been used programmatically?

Page 5, line 2 from bottom: “an interval of three months” is not clear to me, can you rephrase?

Page 5, last line: ref 14 is almost 500 pages long, so please include the page of the report with this information

Page 6, line 8-9: phrasing of “the study methods was quantitative method through…” is an awkward sentence, see if you can rephrase it.

Page 7, paragraph 1: how many schools did you sample, how many students total? What was the response rate?

Page 7, last line: “and lab technician was used for …” I find this phrasing awkward

Page 8, line 6: does the HemoCue Hb30 require a blood sample? More information on this for the non-expert reader would be helpful.

Page 9, table cell on Knowledge on anemia and WIFAS: what was the median, and what did you do with scores with value exactly equal to it?

Page 10, line 4-5 from bottom: is the season during which the data was collected relevant?

Page 11, table 2: is ethnicity/caste relevant to anemia deficiency in this population? I would be interested to know more about this in the methods and discussion.

Page 12, Table 2 continued: there might be more rows than necessary in this table, consider if you can simplify the rows by grouping some categories or leaving things out.

Page 13, line 3-4: your observation that there is no severe anemia in non-WIFAS schools made me wonder again if there is some systematic bias being introduced by the selection of which schools to implement a WIFAS program.

Page 13, paragraph 2: I would like to see a comparison of distributions here, since the means are so similar but the percent anemic have this important difference. A plot overlaying the histogram of hemoglobin levels in the WIFAS and non-WIFAS schools could do this, or a quantile-quantile plot comparing these distributions even more directly might be appropriate.

Page 13, paragraph 3: the approach to the knowledge questions should be described in the methods section, and when you move it there also double check the values, because it seems like it might be inconsistent, e.g. 19 and 15 items add up to maximum of 12 and 10.

Page 13, paragraph 4: This is unclear, and appears potentially inconsistent. 59.1% is not two times more than 54.2, for example, so I’m probably misunderstanding your claims. 78.4 is not two time more than 46.5, either.

Page 13, paragraph 5: this phrasing is a bit confusing and awkward as well.

Page 13-14: I think I need to know about the functional form of your regression model in the methods section to appreciate the implications of this. Is this a sequence of univariate regression models or a single multivariate regression.

Page 14, paragraph 2: I would appreciate a table of figure showing these coefficients

Page 14, last line: mention table 3 earlier.

Page 17, last line of table: the odds ratio for inadequate dietary diversity seems important, and could perhaps be a larger part of the results and discussion. Was this something that you would have expected a priori? If it is a surprise, that is all the more reason to call attention to it.

Page 18, line 2: you say that this is low cost, but you need to present some costing evidence in the discussion (and describe how you obtained it in the methods) to make this claim in the discussion.

Page 18, line 5: the term “WIFAS not implemented schools” has always been awkward in this paper, but this time particularly confused me. I suggest you use something like “non-WIFAS”.

Page 18, line 7: I don’t think it is appropriate to call this statistically significant without making the argument that it can be treated like a random sample of schools which have been randomly assigned to have WIFAS or not.

Page 18, line 6 from bottom: I would be very interested in a visual representation of the effectiveness of WIFAS from different studies and how the results of this study fit into the existing evidence base.

Page 19, line 1-3: your comment about poor compliance made me wonder if your analysis is among people who were treated or among everyone in the school that is offering the treatment?

Page 19, line 4-7: which paper are you referring to here?

Page 19, line 10: mentioning that a different study ran for 6 months made me wonder how long was this WIFAS program running before your study was conducted?

Page 19, line 10 from bottom: “This would result adolescent comply” is awkward phrasing. What was the compliance rate in the WIFAS schools?

Page 19, final paragraph: there is some awkward phrasing in the second sentence here.

Page 20, final paragraph: there are a few things in passing here that I would like to know more about in your results section, such as what side effects were reported and how frequent were they? And what about the dietary diversity?

Page 20, last line: 12.1 times *more likely*? Or 12.1 times *odds ratio* to be anemic?

Page 21, paragraph 2: some of this should go in the methods section.

Page 22, line 2-3: the conclusion that WIFAS significantly reduces anemia strikes me as too bold without more justification for how you have avoided potential bias from which schools implemented WIFAS and which schools were sampled.

Page 22, line 5: odds reported in results section were 30% higher, not 300% higher!

6. PLOS authors have the option to publish the peer review history of their article (what does this mean?). If published, this will include your full peer review and any attached files.

**Do you want your identity to be public for this peer review?** For information about this choice, including consent withdrawal, please see our Privacy Policy.

Reviewer #1: No

Reviewer #2: No

Reviewer #3: **Yes: **Abraham D. Flaxman

---

## [Editor Report · Decision Letter 1]

7 Nov 2023

Prevalence of anemia and its associated factors among adolescent girls on Weekly Iron Folic Acid supplementation (WIFAS) implemented and non-implemented schools at Tokha municipality, Kathmandu

PGPH-D-23-01329R1

Dear Ms/Mrs. Khanak ,

We are pleased to inform you that your manuscript 'Prevalence of anemia and its associated factors among adolescent girls on Weekly Iron Folic Acid supplementation (WIFAS) implemented and non-implemented schools at Tokha municipality, Kathmandu' has been provisionally accepted for publication in PLOS Global Public Health.

Best regards,

Jitendra Kumar Singh, PhD

Academic Editor